# Co-Targeting the EGFR and PI3K/Akt Pathway to Overcome Therapeutic Resistance in Head and Neck Squamous Cell Carcinoma: What about Autophagy?

**DOI:** 10.3390/cancers14246128

**Published:** 2022-12-12

**Authors:** Hannah Zaryouh, Jinthe Van Loenhout, Marc Peeters, Jan Baptist Vermorken, Filip Lardon, An Wouters

**Affiliations:** 1Center for Oncological Research (CORE), Integrated Personalized & Precision Oncology Network (IPPON), University of Antwerp, 2610 Antwerp, Belgium; 2Department of Medical Oncology, Antwerp University Hospital, 2650 Edegem, Belgium

**Keywords:** HNSCC, EGFR, PI3K/Akt, cetuximab, resistance, autophagy, combination therapy

## Abstract

**Simple Summary:**

Combining EGFR-targeted therapies, such as cetuximab, with a potent inhibitor of the PI3K/Akt pathway may be a novel therapeutic strategy that could potentially overcome/circumvent resistance. Interestingly, all pathways downstream of EGFR play a role in the regulation of the autophagic response and combining EGFR-targeted therapies with PI3K/Akt pathway inhibitors can therefore lead to therapy-induced autophagy. In this short commentary, we discuss that therapy-induced autophagy in these kind of combination strategies might not necessarily be a bad sign, as autophagy can also be a cell death mechanism. We highlight the fact that it remains challenging to elucidate the specific cellular requirements to promote autophagic cell death and illustrate this with recent literature. As autophagy also plays a role in anti-tumor immunity by ensuring the release of antigens, potentially leading to recognition and elimination of the tumor, we believe that it is worth investigating autophagy as an anti-tumor mechanism in HNSCC.

**Abstract:**

Resistance to EGFR-targeted therapy is a major obstacle on the road to effective treatment options for head and neck cancers. During the search for underlying mechanisms and regulators of this resistance, there were several indications that EGFR-targeted therapy resistance is (partially) mediated by aberrant signaling of the PI3K/Akt pathway. Genomic alterations in and/or overexpression of major components of the PI3K/Akt pathway are common in HNSCC tumors. Therefore, downstream effectors of the PI3K/Akt pathway serve as promising targets in the search for novel therapeutic strategies overcoming resistance to EGFR inhibitors. As both the EGFR/Ras/Raf/MAPK and the PI3K/Akt pathway are involved in autophagy, combinations of EGFR and PI3K/Akt pathway inhibitors can induce an autophagic response in tumor cells. This activation of autophagy can be seen as a “double-edge sword”, depending on the cellular context. Autophagy is largely known as a cytoprotective mechanism, but it can also be a mechanism of programmed (autophagic) cell death. The activation of autophagy during anti-cancer treatment is, therefore, not necessarily a bad sign. However, in HNSCC, the role of therapy-induced autophagy as an anti-tumor mechanism is still largely unclear. Further research is warranted to understand the potential of combination treatments targeting both the EGFR and PI3K/Akt pathway.

## 1. Introduction

The epidermal growth factor receptor (EGFR) is overexpressed in the majority of head and neck squamous cell carcinomas (HNSCC) [1]. This triggered the development of multiple anti-EGFR agents as a potential treatment strategy for this disease. Despite initial promising results and clinical implementation of one of the first successfully approved targeted therapies in solid tumor treatment, namely the EGFR-specific antibody cetuximab, intrinsic and acquired resistance often occurs with a negative effect on outcome. In theory, pharmacological blockade of EGFR should result in the inhibition of its major downstream signaling pathways, i.e., (i) the Ras/Raf/mitogen-activated protein kinase (MAPK) pathway; and (ii) the phosphatidylinositol 3-kinase (PI3K)/Akt pathway. However, it is becoming more and more clear that the PI3K/Akt pathway often remains activated in patients who exhibit EGFR-targeted therapy resistance [2,3]. In this light, co-targeting the EGFR and PI3K/Akt pathway might be a promising therapeutic strategy to overcome anti-EGFR therapy resistance in HNSCC.

## 2. Genetic Background of Resistance to Anti-EGFR Therapies: Focusing on the PI3K/Akt Pathway

Despite anti-EGFR therapy, the sustained activation of the PI3K/Akt pathway might be explained by looking at the genetic background of the resistant tumor. The presence of activating mutations in genes that lead to the overexpression and/or sustained activation of key mediators of the PI3K/Akt pathway might be involved in the development of resistance [4,5]. Interestingly, the PI3K/Akt pathway is one of the most frequently mutated pathways in HNSCC [6]. Genetic alterations in one of the major components of this pathway are seen in 66% of HNSCC patients [7]. On the other hand, as an increasing proportion of HNSCC are human papilloma virus (HPV) positive, it is worth mentioning that HPV infection can also lead to aberrant activation of the PI3K/Akt pathway. The HPV oncoproteins E6 and E7 have been shown to activate major components of the pathway [8,9], and might therefore also play a role in EGFR-targeted therapy resistance.

According to the TCGA dataset, mutations in the *PIK3CA* gene, which encodes for the catalytic p110 subunit of PI3K, can be found in 21% of HNSCC patients and are common in both HPV-positive and HPV-negative HNSCC [7]. In vitro studies have also shown that cell lines that display activating *PIK3CA* mutations were characterized by an inadequate response to cetuximab treatment, suggesting a role of *PIK3CA* mutational status in cetuximab resistance.

Additionally, the PI3K/Akt pathway is negatively regulated by the tumor suppressor phosphatase and tensing homolog (PTEN), which dephosphorylates PIP_3_, thereby terminating the signaling cascade. Loss of this protein results in a release of the internal breaks on the pathway and is often associated with more aggressive tumors. In HNSCC, low or complete loss of PTEN expression is observed in 10–30% of the patients, regardless of the HPV status [6,7,10,11,12]. It has been shown that PTEN loss may reduce the effectiveness of multiple EGFR inhibitors in HNSCC [13,14]. In addition, there are several indications that it might serve as a predictive biomarker for anti-EGFR therapy response [15].

Mutations in genes encoding for the last two major downstream effector molecules of the PI3K/Akt pathway (Akt and mTOR) are almost non-existing, whereas significant overexpression of these proteins is more frequent [16]. In this regard, increased levels of phospho-Akt following cetuximab treatment [17] and elevated mTOR activity [18] have been reported in HNSCC resistance studies, although their precise role in anti-EGFR resistance remains largely unclear.

## 3. Co-Targeting EGFR and PI3K/Akt Pathway: Match Made in Heaven or Not?

As mentioned above, HNSCC tumors resistant to EGFR inhibitors are often characterized by genetic changes in major players of the PI3K/Akt pathway. Therefore, simultaneous targeting of the EGFR and the PI3K/Akt pathway seems to be a logical step in order to overcome anti-EGFR therapy resistance. Indeed, multiple (pre)clinical studies have demonstrated promising results, showing that combining EGFR and PI3K pathway inhibitors often leads to superior anti-tumor effects compared to either treatment alone. Several underlying mechanisms could be responsible for the effect observed in these combination treatments and one of them is the activation of autophagy.

Autophagy is an evolutionary conserved catabolic process in which cells sequestrate, degrade and recycle their own intracellular contents, such as organelles and proteins. It is tightly linked with metabolism, as autophagy may occur as a response to nutrient deprivation or damaged proteins/organelles. Autophagy is therefore mainly known as a protective mechanism against cellular stress, with the goal to provide the cell with the necessary nutrients in order to survive. In the context of cancer, studies have shown that cancer cells can survive and sustain microenvironmental stress by upregulating autophagy, leading to enhanced growth and aggressive potential of the tumor [19,20,21]. Particularly the ability of autophagy to provide the tumor cell with metabolic energy sources can lead to metabolic plasticity and increased survival [22]. However, autophagy can also be a mechanism of programmed cell death (autophagic cell death, ACD), although there is still controversy over the definition or even existence of ACD [23,24]. In general, this term is used when the cytoplasm of cells demonstrates massive vacuolization during the process of cell death [25]. However, it has been heavily debated whether there is a direct relationship between autophagy and cell death. Is it really cell death caused by the activation of autophagy or is it a type of cell death that is accompanied by autophagy? Multiple developmental studies in *Drosophila melanogaster* have shown that autophagy is (at least partially) involved in a type of cell death [26,27,28,29]. In addition, loss of function of genes that are known to be crucial for the induction (initiation steps) of autophagy has been shown to lead to failure of cell death induction after exposure to several lethal inducers [30]. These observations play in favor of the existence of ACD (at least in specific experimental settings). However, more frequently than not, inhibition of autophagy by pharmacological and/or genetic manipulation leads to accelerated cell death, instead of cell death prevention [31,32]. Therefore, some researchers are convinced that the activation of autophagy by dying cells might also just be a failed attempt of the cells to cope with cellular stress [32,33]. Due to the many paradoxical observations and the ongoing debate around ACD, the Nomenclature Committee of Cell Death 2009 proposed the use of the less stringent description of ‘cell death occurring with autophagy’ instead of ‘ACD’ [25].

The specific role of autophagy in HNSCC is still being investigated. While there is evidence that autophagy modulation might be an interesting new therapeutic strategy, it is still unclear whether inhibition or rather exacerbation of autophagy is the best way forward to improve treatment of HNSCC. Most likely, the disease context will determine the required mode of autophagy modulation [34], as one therapy that fits all is very unlikely in a heterogeneous cancer type such as HNSCC.

All signaling pathways downstream of EGFR are involved in the regulation of autophagy. In this regard, the tyrosine kinase domain of EGFR itself can bind to Beclin1, which leads to phosphorylation of the tyrosine residues of Beclin1, resulting in the formation of Beclin1 homodimers. This triggers the removal of VPS34 from the Beclin1-VPS34 complex, followed by binding of two autophagy suppressors, Rubicon and Bcl2 to Beclin1 (Figure 1) [35,36,37,38]. Consequently, the rearrangement of the Beclin1 complex induced by EGFR activation decreases VPS34 activity and hereby inhibits autophagy [39,40,41]. On the other hand, EGFR-mediated Ras/Raf/MAPK signaling promotes the autophagic response through serine phosphorylation of Beclin1, while signaling through the PI3K/Akt pathway suppresses autophagy by activating mTOR, which is a well-known inhibitor of autophagy [41]. More specifically, activation of mTORC1 leads to phosphorylation of the UNC-51-like autophagy-activating kinase 1 (ULK1) complex, consisting of ULK1, autophagy-related protein 13 (ATG13), and the focal adhesion kinase family interacting protein of 200 kDa (FIP200) [42,43,44]. This deactivating phosphorylation of the complex causes an inhibition of autophagy, since an active ULK1 complex has been shown to be crucial for the initiation of autophagy [45]. In addition, Akt is also able to phosphorylate ULK1 [46] and Beclin1 [47], thereby negatively regulating autophagy on different substrates from a more upstream level in the signaling pathway. When mTORC1 or Akt are inactive (for example due to a PI3K/Akt pathway inhibitor), the inhibitory sites of the ULK1 complex can become dephosphorylated by phosphatases [48,49] followed by autophosphorylation of ULK1 at Thr180 [46]. In turn, active ULK1 phosphorylates ATG13, FIP200 and ATG101 [44,45,50], rendering the complex in its fully activated state (Figure 1). After translocation of the active ULK1 complex to the endoplasmic reticulum (ER), autophagy is initiated. Interestingly, the above-described mechanism is the best characterized way in which mTORC1 inhibits autophagy, but mTORC1 also regulates the class III PI3K (PI3KC3) complex, containing multiple autophagic proteins, such as Beclin1, ATG14 and VPS34 (Figure 1) [51]. While activity of the ULK1 complex is necessary for the initiation, the activity of the PI3KC3 complex is necessary for the nucleation step in the autophagy process [52]. Once ULK1 is active, the PI3KC3 complex is recruited to the ER and further activated by phosphorylation of Beclin1 at Ser15 and Ser30 [53,54], whereas active Akt and mTORC1 inhibit its activity by inhibitory phosphorylation of the complex (Figure 1) [52]. The activation and translocation of the PI3KC3 complex result in the initiation of nucleation. During this process, additional ATG proteins are recruited to the phagophore assembly site that are required for further expansion and maturation of the phagophore [55,56]. Once the autophagosome has been formed, which is the fully assembled form of the phagophore, lysosome fusion and subsequently degradation/recycling of intracellular components will occur (Figure 1) [55,56].

In these ways, the EGFR and PI3K/Akt pathways prominently play a role in the activation or inhibition of autophagy. Therefore, blocking both EGFR and the PI3K/Akt pathway as a therapeutic strategy to overcome resistance of EGFR-targeted agents in HNSCC may result in an increased activation of the autophagic response, leading to either cell survival or cell death. Which path will be chosen (cell survival or cell death) is uncertain and depends on multiple factors, such as cell type, genetic background and prevailing microenvironment; for example, radiation-induced autophagy was shown to be cytoprotective in the p53 wild-type HNSCC HN30 cell line, while it was nonprotective in the p53 mutant HN6 cell line [57]. Similarly, irradiating the OC3 HNSCC cell line resulted in the activation of autophagy, and also cell death. However, it remains unclear whether ACD was activated or autophagy appeared as a reaction to radiation-induced cellular stress [58]. As illustrated by the latter examples, it remains challenging to elucidate the specific cellular requirements to promote ACD.

Activation of autophagy during anti-EGFR therapy is commonly seen as a bad sign, as there are several indications that the autophagic response might be a treatment escape mechanism and thus involved in the development of therapeutic resistance. In this regard, cetuximab resistant colorectal and HNSCC cells could be sensitized to cetuximab by inhibition of autophagy, suggesting autophagy to be a protective mechanism in these cells, mediating cetuximab resistance [59,60]. Activation of autophagy by induction of oxidative stress was shown to diminish the efficacy of erlotinib in HNSCC cell lines [61]. Likewise, PI3K/Akt pathway inhibitors have been reported to induce protective autophagy, supporting the unwanted cell survival of tumor cells. In this context, combining PI3K/Akt inhibitors with autophagy inhibitors, such as (hydroxy)chloroquine, has demonstrated superior anti-proliferative effects in HNSCC cell lines compared to PI3K/Akt inhibitors alone [62]. Chloroquine inhibits autophagy by preventing the fusion of autophagosomes with lysosomes, resulting in the accumulation of a large number of autophagosomes in the cytoplasm, which eventually might lead to cell death [63,64]. Moreover, in other cancer types, pro-survival autophagy has been demonstrated when using PI3K/Akt pathway inhibitors, even in combination with anti-EGFR therapy. Namely, the study of Bokobza et al. showed that the combination of gefitinib with and Akt inhibitor induced a protective mechanism in non-small cell lung cancers cells carrying an EGFR mutation. Although the combination was synergistic, adding chloroquine to the combination led to significantly enhanced tumor cell death, suggesting that the therapy-induced autophagy acted as a compensatory pro-survival mechanism [65].

The latter studies do not quite advocate for the dual targeting of EGFR and PI3K/Akt pathway inhibitors as a promising strategy to overcome resistance in HNSCC. However, as mentioned previously, autophagy can also be a mechanism of cell death, although this cytotoxic element remains largely unclear and warrants further investigation. In addition, autophagy plays a role in anti-tumor immunity, as it ensures the release of antigens, potentially leading to tumor recognition and elimination [66]. Therefore, it is worth investigating autophagy as an anti-tumor mechanism in HNSCC. In this regard, deguelin, an autophagy inducer and Akt signaling inhibitor, has been shown to promote HNSCC cell death and to sensitize HNSCC cells to 5-fluorouracil, indicating that the activation of autophagy contributed to HNSCC cell death [67]. More specifically for treatments targeting both EGFR and the PI3K/Akt pathway, the presence of autophagy has already been suggested to be an underlying cell death mechanism, rather than a cell survival mechanism. In this context, the study of Li et al. has demonstrated that combining cetuximab with the mTOR inhibitor rapamycin leads to a synergistic effect in A431 vulvar carcinoma cells [59]. Cetuximab alone induces mainly a cell cycle arrest and weak apoptosis in these cells. Inhibition of autophagy by knockdown of *Atg5* largely abolished the added value of rapamycin to the treatment, strongly indicating that autophagy served as a cell death mechanism. In addition, HNSCC FaDu and HN5 cell lines that only show a growth arrest without apoptosis following cetuximab treatment, also benefited from the combination of rapamycin plus cetuximab, but this was antagonized by the addition of the autophagy inhibitor chloroquine. This suggests that the addition of rapamycin to cetuximab also stimulated ACD in these HNSCC cells. The authors conclude that ACD following treatment with cetuximab and rapamycin is dependent on the response of the cells to cetuximab alone. The activation of the autophagy pathway led to ACD only in cells that responded to cetuximab alone with weak apoptosis or a growth arrest. Interestingly, cells that show strong apoptosis after cetuximab treatment benefited from a combination of cetuximab and an autophagy inhibitor, as autophagy acted as a cytoprotective mechanism in these cells [59]. These results highlight again that the final effect induced by autophagy (cell survival or cell death) is highly dependent on the cellular context.

The study of D’Amato et al. also demonstrated that there was a highly synergistic effect when cetuximab was combined with the dual PI3K/mTOR inhibitor PKI-587 in both cetuximab sensitive and resistant cell lines. Interestingly, cetuximab sensitive cell lines were characterized by activation of apoptosis, compared to cetuximab resistant cell lines that showed activation of autophagy. More specifically, resistant cell lines had an increased expression of Beclin1 and a decreased expression of p62 following combination treatment. Since a significant inhibition of cancer cell proliferation was observed in all cell lines, the authors concluded that the combination of PKI-587 and cetuximab induced apoptotic cell death in sensitive cell lines and a different type of cell death, i.e., ACD, in resistant cell lines [68]. Thus, in the study of D’Amato et al., the observed activation of autophagy was not mediating therapy resistance, as is often described in the literature [60,61,62], but rather overcoming it as an indirect effect of the combination regimen.

Although the two above-mentioned studies support the beneficial effect of therapy-induced autophagy, it is worth mentioning that more recent literature, showing that autophagy can act as a cell death mechanism when combining anti-EGFR and PI3K/Akt pathway inhibitors, is lacking. This may be due to the fact that the majority of studies describing such combinations in HNSCC simply did not investigate whether the observed synergistic effect was accompanied by activation of autophagy. To date, this remains a question to be answered. As we have discussed studies that demonstrated the presence of autophagy as a cell survival mechanism, but also found evidence that supports autophagy as a cell death mechanism after combined targeting of the EGFR and PI3K/Akt pathways, we sincerely believe that the specific cellular (and maybe even experimental) context plays a pivotal role in deciding the final path of autophagy (cell survival or cell death). In this regard, the degree of autophagy that is induced by the treatment, the cancer type, the type of inhibitors (tyrosine kinase inhibitors or monoclonal antibodies) used, the cell’s metabolic status, as well as the genetic characteristics of the cells, are very likely influencing the path that will be chosen. In this light, the study of Li et al. clearly showed that the way cells respond to cetuximab therapy alone influences whether it would be beneficial to combine it with a PI3K/Akt pathway inhibitor to activate autophagy. The latter is a cell intrinsic property that is shaped by the cell’s dependence on signaling through the EGFR pathway for growth and survival [59]. As there is a clear gap of knowledge in this area, we are convinced that future investigations should focus on further unravelling the role of therapy-induced autophagy and especially the cellular requirements to induce ACD in HNSCC.

## 4. Conclusions

Aberrant signaling of the PI3K/Akt pathway is involved in resistance to EGFR-targeted therapies. Genomic alterations in and/or overexpression of the major components of the PI3K/Akt pathway are common in both HPV-positive and HPV-negative HNSCC tumors. Therefore, downstream effectors of the PI3K/Akt pathway serve as promising targets in the search for novel therapeutic strategies that are able to overcome resistance to EGFR inhibitors. EGFR and PI3K/Akt pathway inhibitors or combinations thereof can induce an autophagic response in tumor cells. The activation of autophagy during anti-cancer therapy can be seen as a “double-edge sword”, depending on the cellular context. The role of therapy-induced autophagy following dual targeting of the EGFR and PI3K/Akt pathway as an anti-tumor mechanism is still largely unclear in HNSCC. Further research is warranted to fully understand the potential of combination treatments targeting both the EGFR and PI3K/Akt pathway.

## Figures and Tables

**Figure 1 cancers-14-06128-f001:**
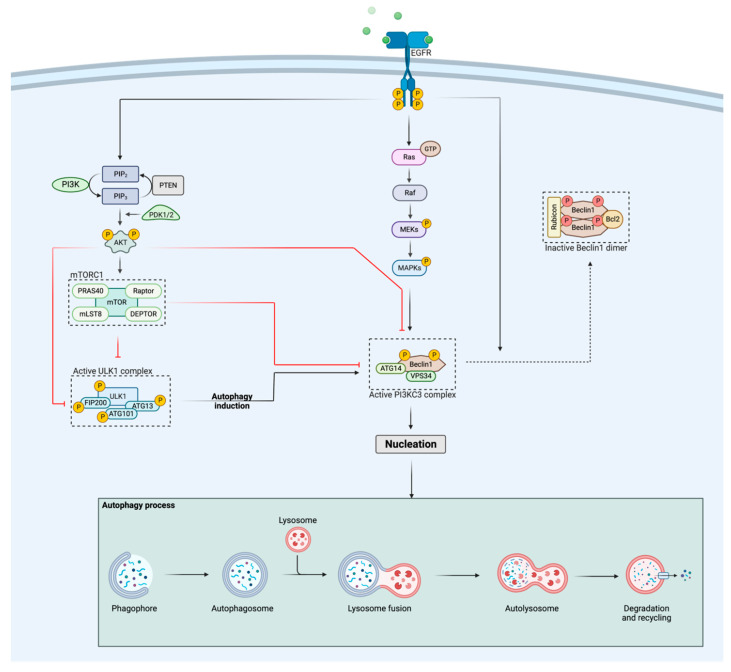
Schematic representation of the regulation of autophagy by the EGFR and PI3K/Akt pathways. Activation of EGFR stimulates the PI3K/Akt pathway, which in turn negatively regulates autophagy activation via different ways: (i) Akt and mTORC1 inhibit the PI3KC3 complex through phosphorylation and (ii) Akt and mTORC1 suppress the activity of the ULK1 complex, hereby preventing autophagy initiation. The Ras/Raf/MAPK pathway, activated by EGFR, positively regulates the autophagic response through phosphorylation of Beclin1, whereas activated EGFR itself negatively regulates autophagy by binding to Beclin1, followed by homodimerization of Beclin1 and the binding of two autophagy suppressors, Rubicon and Bcl2, rendering the complex inactive. When both the ULK1 and PI3KC3 complex are released from their inhibitory signals, nucleation will be initiated, leading to further expansion of the phagophore. The figure was created with BioRender.com (accessed on 23 October 2022).

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
