# Peer review of "Co-Targeting the EGFR and PI3K/Akt Pathway to Overcome Therapeutic Resistance in Head and Neck Squamous Cell Carcinoma: What about Autophagy?"

_cancers, 2022, doi:10.3390/cancers14246128_

Round 1

Reviewer 1 Report

The commentary titled “Co-targeting the EGFR and PI3K/Akt pathway to overcome therapeutic resistance in head and neck squamous cell carcinoma: what about autophagy?” focusses on the role of autophagy in mitigating resistance to EFGR therapy using a PI3K/Akt inhibitor. This is well-written commentary article and highlights the controversial role of autophagy in treatment resistance. EGRF-Targeted therapy (cetuximab) is approved for the treatment of HNSCC. However, patients often develop resistance to EGRF-Targeted therapy thereby limiting its efficacy. Cetuximab resistance is accompanied by the activation of it’s downstream PI3K/Akt pathway.  Therefore, combining EGFR-targeted therapy with mTOR inhibitor (mTORi), might be a effective strategy to overcome anti-EGFR therapy resistance in HNSCC. Recent data suggest combining EGFR and PI3K pathway inhibitors exhibited superior efficacy than either treatment alone, the induction of autophagy might make cetuximab resistance tumors sensitive to combination treatment.

Minor comments:

1.      Induction of autophagy and activation of autophagy are not the same.

The author used the term induction of autophagy and activation of autophagy equivocally, however induction of autophagy is the initiation steps of autophagy, which may or may not results in activation of autophagy. These terminologies needs to be clear and distinctive in the context of the review.  It would add value to the commentary if the author could expand on the cellular context ( mutations particularly) in which autophagy promotes cell survival or cell death. Again, the author needs to clarify induction or activation of autophagy. Induction (initiation) of autophagy might not results in activation of autophagy if later stages are inhibited.   

2.      Increased or decreased level of p62 is a marker for activation of autophagy.

The authors mention activation of autophagy by increased expression of p62 is associated with cetuximab resistance; clearly increased p62 level is a marker for autophagy activation. However in D’ Amato et al’s study decreased expression of p62 was linked to autophagy induction. The author needs to clarify whether increased or decreased level of p62 is linked with activation of autophagy. In general decreased level of p62 is known as a marker for activation of autophagy.

Combining autophagy inhibitor (chloroquine) with the PI3K/Akt pathway inhibitors study needs to be explained clearly in the context of autophagy. Chloroquine inhibits autophagy by inhibiting the formation of autolysosomes but it does not prevents the induction (initiation) of autophagy. Therefore combining PI3K/Akt pathway inhibitors with chloroquine might results in defective autophagy vesicles accumulated in the cells leading to cell death.

Reviewer 2 Report

 The authors discuss on the interaction of tho different cellular pathways involved in the development of H&N cancer.  EGF-R target therapy and PI3K/atk patways may represent an important driver for H&K therapy.

The mauscript is well written

Author Response

Comments of reviewer 2:
We would like to thank reviewer 2 for the positive evaluation of the manuscript.

Reviewer 3 Report

In this Commentary the authors speculate on the possible role of autophagy in overcoming/preventing resistance to EGFR-targeted therapy by combining EGFR inhibitors with inhibitors of the PI3K/AKT pathway in head and neck squamous cell carcinoma (HNSCC). Considering that EGFR is overexpressed in the majority of HNSCC, the anti-EGFR monoclonal antibody cetuximab was approved (in combination with radiation therapy or chemotherapy) in the treatment of this type of cancer; however its efficacy is limited due to drug-resistance mechanisms. Since the PI3K-AKT pathway is often responsible for resistance to EGFR-targeted therapies in HNSCC, the authors suggest that co-targeting the two pathways might represent a valid therapeutic option to counteract drug resistance, and cite several preclinical studies supporting this concept. Actually, the main focus of this article is the potential role of autophagy, induced by dual inhibition of EGFR-PI3K/AKT pathways, in the success of this combined therapy. Notably, autophagy is known to lead to either cell survival or cell death depending on the cellular context. The authors make the point here that the simultaneous inhibition of EGFR and PI3K/AKT might promote a form of autophagy leading to cell death, thereby preventing resistance to EGFR targeted therapies.

Overall the Commentary is very well written and the background literature review appears to be accurate. The working hypothesis illustrated by the Authors is original and interesting, although usually autophagy induced by EGFR inhibition is rather considered a mechanism of resistance to EGFR targeted therapy (or “a bad sign”, as the authors write). AKT inhibition, as well, has been shown to induce pro-survival autophagy response, as a mechanism to escape cytostatic drug effects. The novel concept put forward by the authors here is that most previous studies have not assayed the effects of dual inhibition of EGFR and AKT on autophagy regulation and cancer cell viability. Actually, in support to their hypothesis of a beneficial effect of such combined therapy, the authors cite a single study (D’Amato et al, 2014), in which co-targeting of EGFR and PI3K/mTOR caused decreased proliferation of HNSCC cells resistant to cetuximab, through the induction of autophagy-related cell death. It is worth mentioning, however, that in another published study the combination of chloroquine (an inhibitor of autophagy) with EGFR and AKT inhibitors led to blockade of pro-survival autophagy and increased cell death (Bokobza  et al, 2014; PMID: 24946858). In other words, in this case, a pro-survival autophagy mechanism was used by EGFR/AKT-inhibited cancer cells to survive the therapy, opposite to what the authors envisage in this Commentary. It would therefore be appropriate to discuss also such contrasting evidence in the paper. Notably, in the Bobobza study, a different type of cancer (lung versus head and neck) and a different type of EGFR inhibitor (kinase inhibitor versus monoclonal antibody) were investigated; thus, the authors could discuss these differences and underscore the fact that their working hypothesis might be strictly dependent on the cellular context (e.g., type of cancer, type of inhibitor).

Reviewer 4 Report

The manuscript described the molecular back ground of the nature of the chemoresistance of HNSCC through EGFR and PI3K/AKT signaling in the context of autophagy. The referee feels that the topic is important in both clinical as well as basic cancer biological point of view.

The referee feels that the connection of cancer metabolism may be missing in this manuscript (see belo). The referee feel that some numbers of important references are missing, which authors need attention.

#1. Many important references  relevant to this manuscript appear to be are missing:

Green, D. R., Galluzzi, L. & Kroemer, G. Mitochondria and the autophagyinflammation- cell death axis in organismal aging. Science 333, 1109–1112 (2011).

 Kroemer, G. et al. Classification of cell death: recommendations of the

Nomenclature Committee on Cell Death 2009. Cell Death Differ. 16, 3–11

(2009).

Green, D. R. The coming decade of cell death research: five riddles. Cell 177,

1094–1107 (2019).

#2. Beclin1 Beclin-1 and ULK1 (Atg1) are both involved in autophagosome

formation and are direct substrates of Akt.

Bach, M., Larance, M., James, D. E. & Ramm, G. The serine/threonine kinase

ULK1 is a target of multiple phosphorylation events. Biochem. J. 440,

283–291 (2011).

Wang, R. C. et al. Akt-mediated regulation of autophagy and tumorigenesis

through Beclin 1 phosphorylation. Science 338, 956–959 (2012).

Beclin1 and Bcl-2 are both known to be involed in the regulation of autophagy:

Chipuk, J. E.,Moldoveanu, T., Llambi, F., Parsons, M. J. & Green, D. R. The BCL-2

family reunion. Mol. Cell 37, 299–310 (2010).

Levine, B., Sinha, S. & Kroemer, G. Bcl-2 family members: dual regulators of

apoptosis and autophagy. Autophagy 4, 600–606 (2008).

#3. Autophagy is known to be involved in the cancer metabolism by Drs White and others:

Green DR, Galluzzi L, Kroemer G. Cell biology. Metabolic control of cell death. Science.;345:1250256 ( 2014).

Santana-Codina, N.,Mancias, J.D. & Kimmelman, A. C. The roles of autophagy

in cancer. Annu. Rev. Cancer Biol. 1, 19–39 (2017).

Kimmelman, A. C. & White, E. Autophagy and tumor metabolism. Cell Metab.

25, 1037–1043 (2017).

White, E. The role for autophagy in cancer. J. Clin. Invest. 125, 42–46 (2015).

#4. Fig 1. Autophagy process need to be shown more precisely by showing how the individual ATG molecules involved. Since schematic view of the current manuscript only showed the general vision of macro autophagy, which may or may not be relevant for the molecular process of autophagy.  How mechanistically“ nucleation” bridge the EGFR signal and macroautophagy is not at all clear, which authors should clearly describe along with the appropriate reference(s).

Round 2

Reviewer 3 Report

In this revised version of the article, in response to my comments, the Authors have cited a second paper supporting their working hypothesis (Li et al; ref. 59). As mentioned before, the idea of targeting EGFR and the AKT pathway in HNSCC in order to induce autophagy and  cell death (instead of cell survival), to prevent EGFR therapy resistance, is quite interesting. However, it is worth to mention that both papers supporting the model proposed by the authors (D’Amato et al, and Li et al) are quite old (published in 2014 and 2010, respectively). As far as I can see, there are no recent papers, in the last 12-8 years, supporting this working hypothesis, which seems to weaken the relevance of the proposal. It would be appropriate to underscore this lack of recent supporting literature in the manuscript.

About another discussed aspect, the authors added a paragraph (242-254), in which they appropriately specify that the cell fate (autophagy-induced cell death or cell survival) is affected by the specific cellular and experimental context.
